# Reconstruction of genetically identified neurons imaged by serial-section electron microscopy

Maximilian Joesch[1,2], David Mankus[1,2], Masahito Yamagata[1,2], Ali Shahbazi[3], Richard Schalek[1,2], Adi Suissa-Peleg[4], Markus Meister[5], Jeff W Lichtman[1,2], Walter J Scheirer[3], Joshua R Sanes[1,2]*

[1]Center for Brain Science, Harvard University, Cambridge, United States; [2]Department of Molecular and Cellular Biology, Harvard University, Cambridge, United States; [3]University of Notre Dame, Notre Dame, United States; [4]School of Engineering and Applied Sciences, Harvard University, Cambridge, United States; [5]Division of Biology, California Institute of Technology, Pasadena, United States

**Abstract** Resolving patterns of synaptic connectivity in neural circuits currently requires serial section electron microscopy. However, complete circuit reconstruction is prohibitively slow and may not be necessary for many purposes such as comparing neuronal structure and connectivity among multiple animals. Here, we present an alternative strategy, targeted reconstruction of specific neuronal types. We used viral vectors to deliver peroxidase derivatives, which catalyze production of an electron-dense tracer, to genetically identify neurons, and developed a protocol that enhances the electron-density of the labeled cells while retaining the quality of the ultrastructure. The high contrast of the marked neurons enabled two innovations that speed data acquisition: targeted high-resolution reimaging of regions selected from rapidly-acquired lower resolution reconstruction, and an unsupervised segmentation algorithm. This pipeline reduces imaging and reconstruction times by two orders of magnitude, facilitating directed inquiry of circuit motifs.

*For correspondence: sanesj@mcb.harvard.edu

**Competing interests:** The authors declare that no competing interests exist.

## Introduction

Relating neural circuits to the computations they perform requires mapping patterns of structural and functional connectivity among neurons. Innovative light microscopic methods such as GRASP, trans-synaptic viral tracing and super-resolution imaging enable visualization of synapses made on or by identified neurons (*Wickersham and Feinberg, 2012*; *Tønnesen and Nägerl, 2013*). At present, however, only electron microscopy (EM) provides sufficient resolution to visualize the complete complement of synapses that neurons form and receive. Indeed, large-scale reconstructions from serial sections have provided deep insights into neuronal circuit principles (*Briggman et al., 2011*; *Bock et al., 2011*; *Takemura et al., 2013*; *Morgan et al., 2016*) . The optimal strategy is to collect serial sections containing all circuit elements and image them at nanometer resolution. Processes in the imaged volumes are then segmented to reconstruct the neurons (or parts of neurons) they contain. Advances in sectioning, imaging and segmentation methods make so-called 'saturated' reconstruction of volumes around 1000 $\mu m^3$ feasible (*Kasthuri et al., 2015*). Even these modest volumes remain challenging, however, and when multiple samples must be compared -e.g., controls vs. mutant or treated vs. untreated animals - this approach is currently out of reach (*Plaza et al., 2014*).

An attractive alternative is 'sparse' reconstruction of specific cells within a fully imaged volume. For example, neuronal activity can be monitored using calcium indicators, then neurons with

**eLife digest** Neurons connect with each other to form complex circuits that underlie mental activities. Mapping these connections to obtain a so-called wiring diagram is an essential step in learning how the brain works. The only way to do this precisely enough is by using electron microscopy. However, this technique is so time-consuming that thousands of hours of work are typically required to image even the smallest of tissue samples.

Electron microscopes fire beams of electrons at tissue samples, and detect the scattering of the electrons. Stains are used to make specific neurons less permeable to electrons, or more "electron dense". Labeled cells scatter more electrons, which increases the contrast of the images. In an approach called serial-section electron microscopy, a tissue sample is first cut into extremely thin sections. These are imaged individually, and the images are then pieced together to reconstruct the sample.

Joesch et al. have now developed a new procedure – named ARTEMIS – that uses a combination of multiple techniques to speed up the mapping of neurons and their connections. ARTEMIS makes use of genetic engineering, serial-scanning electron microscopy, an enhanced chemical staining procedure and a new image processing approach. First, gene technology is used to selectively stain specific types of neurons in mice and flies. Then, a tissue sample is collected and treated with a chemical that enhances the electron density of the stained neurons, without disrupting the tissue's structure. Next, a labeled target neuron is imaged at relatively low resolution to reveal its overall structure. Small areas of that neuron are then re-imaged at higher resolution to map the connections between neurons. Lastly, an algorithm pieces together the individual images to produce a reconstruction of the cell.

This pipeline of steps reduces the time required to map the shapes and connectivity of neurons with electron microscopy by some two orders of magnitude. This should enable neuroscientists to obtain more rapid insights into the roles of specific neural circuits in the brains of healthy animals, and to identify cases where this wiring goes awry and leads to disease.

particular patterns of activity can be relocated in thin sections and reconstructed (*Briggman et al., 2011*; *Bock et al., 2011*). This method is, however, technically demanding and infeasible in many tissues. We therefore devised an alternative approach to sparse reconstruction that relies on marking specific cells with an electron-dense tracer. We then exploit the high contrast provided by the tracer to speed up imaging and reconstruction, which are currently the rate-limiting steps in connectomic analysis. Our pipeline includes the following series of steps: (a) tagging a specific cell type with a genetically encoded EM tracer, (b) enhancing the electron-density of the stain without compromising ultrastructure of the surrounding tissue, (c) imaging the cell rapidly at relatively low resolution, (d) re-imaging selected small volumes at higher resolution to map connectivity and (e) segmenting the cell using a novel algorithm that is reliable, fast and does not require computationally intense pre-training. Together, the gains from targeted reimaging and unsupervised segmentation decrease the time required for reconstruction by over two orders of magnitude. We call the method ARTEMIS for Assisted Reconstruction Technique for Electron Microscopic Interrogation of Structure.

## Results and discussion

A classical ultrastructural tracer is horseradish peroxidase (HRP), which catalyzes the formation of a 3,3′-diaminobenzidine (DAB) polymer; the polymer binds osmium and is thereby rendered electron-dense. Recombinant HRP is enzymatically inactive in the cytosol because it fails to form disulfide bonds or become glycosylated, but this limitation can be overcome by directing the protein to topologically extracellular compartments such as vesicles (*Li et al., 2010*; *Atasoy et al., 2014*; *Schikorski et al., 2007*). We therefore generated an HRP variant that was codon-optimized, mutated to increase activity, and fused to an endoplasmic reticulum targeting sequence (erHRP). We also tested derivatives of plant ascorbate peroxidases called APX and APEX2, which are active in the cytosol (*Martell et al., 2012*; *Lam et al., 2015*). Initial studies using cultured HEK293 cells confirmed

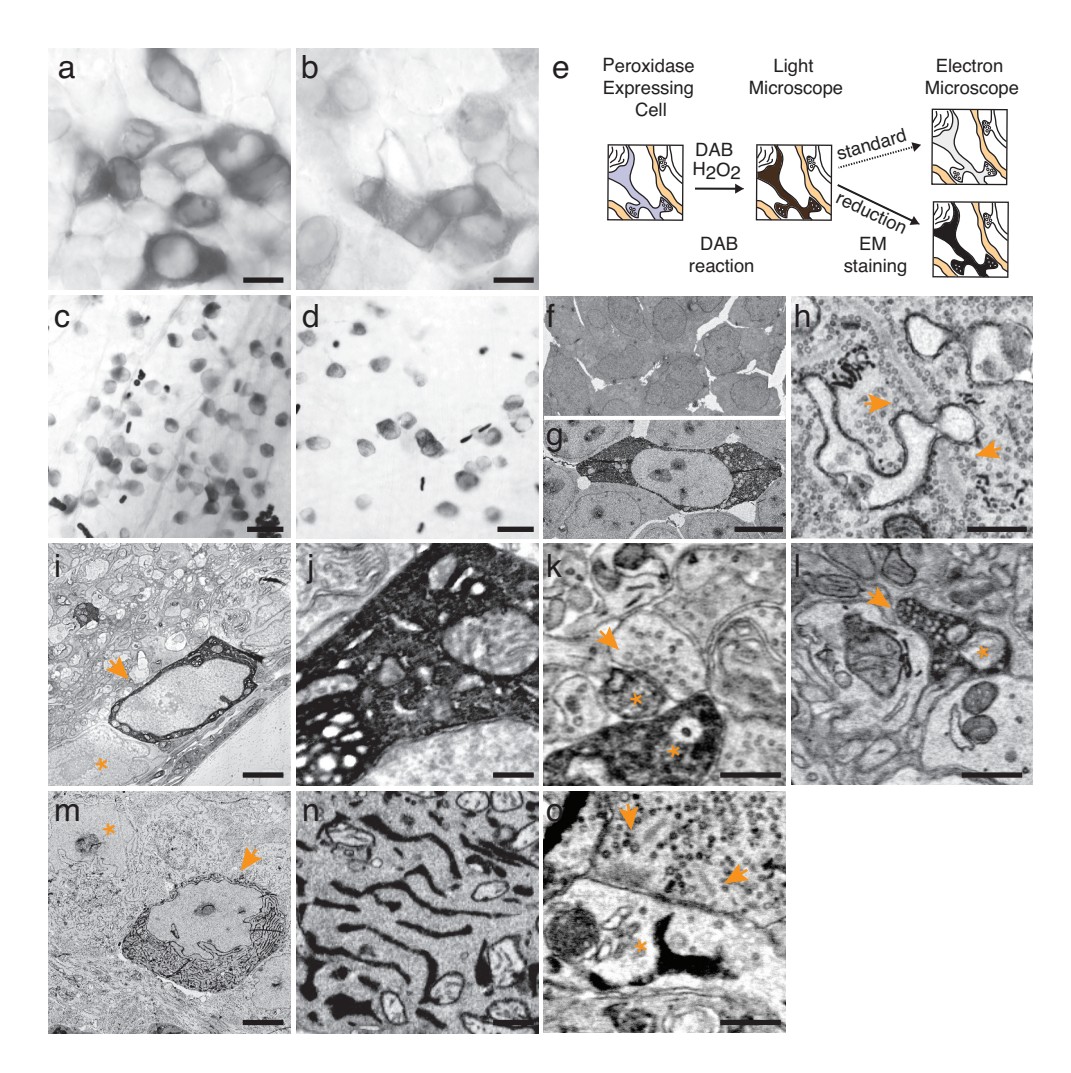

**Figure 1.** Enhanced staining of genetically encoded tags for EM. (**a–b**) Bright-field images of HEK-cells rendered photon-dense by DAB polymerization, catalyzed either by APEX2 tagged with a nuclear export signal (APEX2NES) (**a**) or endoplasmic reticulum tagged HRP (erHRP) (**b**). (**c–d**) Bright-field images of direction-selective retinal ganglion cells (ooDSGC) expressing APEX2NES (**c**) or erHRP (**d**) and rendered photon-dense as in (**a**). (**e**) Enhanced staining strategy. (**f–g**) EM-micrographs of HEK-cells rendered electron-dense after DAB-polymerization as in (**a**). A standard EM staining protocol showed no detectable cytosolic DAB-polymer staining (**f**), whereas the addition of a reduction step dramatically enhanced DAB-polymer staining (**g**). (**h**) Ribbon synapses in the outer plexiform layer; ultrastructure is well preserved after reduction. (**i**) EM-micrograph of an ooDSGC soma (arrowhead) rendered electron dense after tissue reduction next to a non-expressing RGC (asterisk). (**j**) Close-up of cytosolic APEX2 staining. (**k**) Dendritic processes expressing APEX2 (asterisk) contacted by a presynaptic partner (arrowhead) in the inner plexiform layer. (**l**) Axonal long-range projections of an APEX2-expressing ooDSGC in the superior colliculus (asterisk) with a postsynaptic target (arrowhead). (**m**) erHRP-expressing (arrowhead) RGC next to a non-expressing RGC (asterisk). (**n,o**) Close-up of erHRP staining of a J-RGC soma (**n**) and dendrite (**o**) (arrowhead point to presynaptic partners). (c,d,i-l) are from *Cart-cre* mice; (m–o) from *Jam-B-creER* mice. Scale bars: (**a–d**) :25 μm; (**f–g**): 10 μm; **l**, **m** : 5 μm; **h**, **j—l**, **n,o** : 500 nm;

The following figure supplements are available for figure 1:

**Figure supplement 1.** DAB-polymer enhancement.

**Figure supplement 2.** Cytosolic APEX2.

*Figure 1 continued on next page*

*Figure 1 continued*

**Figure supplement 3.** Endoplasmic reticulum tagged HRP (erHRP).
**Figure supplement 4.** Chemical tissue reduction improves contrast-to-noise ratio between membrane and cytosol.
**Figure supplement 5.** Synaptic contacts received and made by peroxidase expressing cells.

that all three constructs generated active peroxidase in the transfected cells (*Figure 1a,b* and data not shown).

For selective expression in molecularly-identified cells, we generated adeno-associated viral (AAV) vectors in which expression of erHRP, APX or APEX2 required Cre-dependent recombination. These were used to infect retinas of transgenic mice in which specific retinal ganglion cell (RGC) types expressed Cre recombinase (ooDSGCs in Cart-*cre* [*Kay et al., 2011*]) or tamoxifen-activated Cre (J-RGCs in JAM-B-*cre*ER [*Kim et al., 2008*; *Joesch and Meister, 2016*]). Two to four weeks after infection, retinas were reacted with DAB and $H_2O_2$ and examined light-microscopically, revealing intense labeling of RGCs (*Figure 1c,d*). Retinas were then processed for EM (see Methods). Unexpectedly, levels of electron-dense precipitate were so low that stained processes could not be traced reliably (*Figure 1—figure supplement 1a,b*). Numerous alterations to balance peroxidase activity and ultrastructural quality failed to improve matters: when ultrastructure was adequately preserved, staining for peroxidase was poor, and when reaction product was adequate, synaptic structures were poorly preserved. This difficulty may have been less apparent in previous studies using injected native HRP, which has substantially higher activity than the recombinant proteins we use.

The reason for the difference between light and electron microscopic results is likely that the opacity and electron density of the DAB polymer arise in different ways: its polycyclic structure renders it photon absorbent, but its electron-density results from redox reactions with osmium (*Bahr, 1954*). We reasoned that radicals produced during the long peroxidase reaction might oxidize the relevant functional groups in the polymer, leaving it photon-absorbent but inert to osmium. If this were true, reduction of functional groups on the polymer could restore reactivity to osmium (*Figure 1e*). We tested this hypothesis in transfected HEK cells. When cells were treated with the protocol we had used for retina, the precipitate was clearly visible by light but not electron microscopy. However, when the HEK cells were treated with a mild reducing agent (5 mM sodium hydrosulfite) between the peroxidase reaction and osmication, they were highly electron-dense (*Figure 1f,g*). This was the case using either conventional osmium staining or an enhanced 'double osmium' staining protocol (rOTO), although the latter showed a slight improvement probably due to the change in the redox state of osmium tetroxide (*Figure 1—figure supplement 1c–f*; see Methods). A similar effect was observed in erythrocytes, in which endogenous heme catalyzes the DAB reaction (*Figure 1—figure supplement 1g–j*).

When the reduction protocol was applied to retinas, we were able to visualize RGCs that had been tagged with APX, APEX2NES (APEX2 fused to a nuclear export signal) or erHRP (*Figure 1i–o*, *Figure 1—figure supplement 2* and *3*). As expected, APX and APEX2NES labeled the cytoplasm diffusely (*Figure 1j*) while erHRP labeled membrane-bound intracellular compartments (*Figure 1n*). Thus they could be used as orthogonal labels, although we have not yet pursued this application. The strength of the signal allowed us to identify small stained dendritic profiles (*Figure 1k,o*) and to view the terminals of RGCs in the superior colliculus, approximately 1 cm from the somata (*Figure 1l*). Remarkably, the reduction protocol actually improved the visualization of the ultrastructure irrespective of peroxidase expression or DAB treatment (*Figure 1h*). Part of the improvement resulted from an increase in the reactivity of membranes to osmium, thereby enhancing membrane-cytoplasm contrast (*Figure 1—figure supplement 4*). This improvement is also visible in the staining strength of synaptic densities (*Figure 1—figure supplement 4a,b*). When we asked blind-to-condition observers to judge the quality of synapses between both conditions, the reduced tissue was selected ~3 times more frequently than unreduced tissue (*Figure 1—figure supplement 4e*). Thus, rather than sacrificing ultrastructure for reactivity or vice versa, this protocol improved both.

For the approach to be useful, it is essential that peroxidase expression does not affect synapse formation and that synaptic partners of peroxidase expressing cells can be identified. When analyzing our datasets, we could not detect any structural differences between tissues expressing either of the peroxidases. The number of synapses counted in peroxidase expressing and control sections was also similar, 0.32 and 0.34 synapses / $\mu m^2$, respectively. To test whether the electron dense precipitate hinders reliable synaptic identification, we characterized the synaptic connections received and made by APEX2 or erHRP expressing retinal ganglion cells (*Figure 1k,o*; *Figure 1— figure supplement 5*). Cells presynaptic to APEX2 or erHRP could be clearly identified based on ultrastructural details (*Figure 1—figure supplement 5a,c*). Postsynaptic partners of APEX2-expressing cells could also be identified. Moreover, synaptic vesicles of APEX2-expressing neurons could be detected in presynaptic terminals because they were unstained, and thereby contrasted with the electron-dense cytosol. However, the endoplasmic reticulum did not regularly extend to axonal terminals, making it difficult to identify postsynaptic partners of erHRP expressing ganglion cells.

To test the generality of the method we expressed APEX fused to GFP in direction-selective tangential cells of *Drosophila melanogaster* (*Joesch et al., 2008*) (*DB331-Gal4 → UAS-APEX-GFP*; see Methods). This driver line expresses mainly in 6 vertically sensitive and 3 horizontally sensitive tangential cells. This expression pattern was apparent using either GFP fluorescence or the polymerized DAB to mark the cells (*Figure 2a,b*). We were also able to visualize the electron dense precipitate of these processes in the fly's optic lobes (*Figure 2c,d*). Although our method was optimized for mammalian tissue, ultrastructural detail was reasonable (*Figure 2e*) and allowed the identification of synaptic contacts made by and onto APEX expressing cells (*Figure 2f–h*).

To implement sparse reconstruction, we used the ATUM (*Hayworth et al., 2014*) (automated tape-collecting ultra-microtome) to serially section retinas containing either APEX2NES-expressing retinal ganglion cells (J-RGCs) or retinal interneurons (starburst amacrine cells (SACs)) labeled using Choline Acetyltransferase-Cre (*Rossi et al., 2011*). Because we could identify small peroxidase-

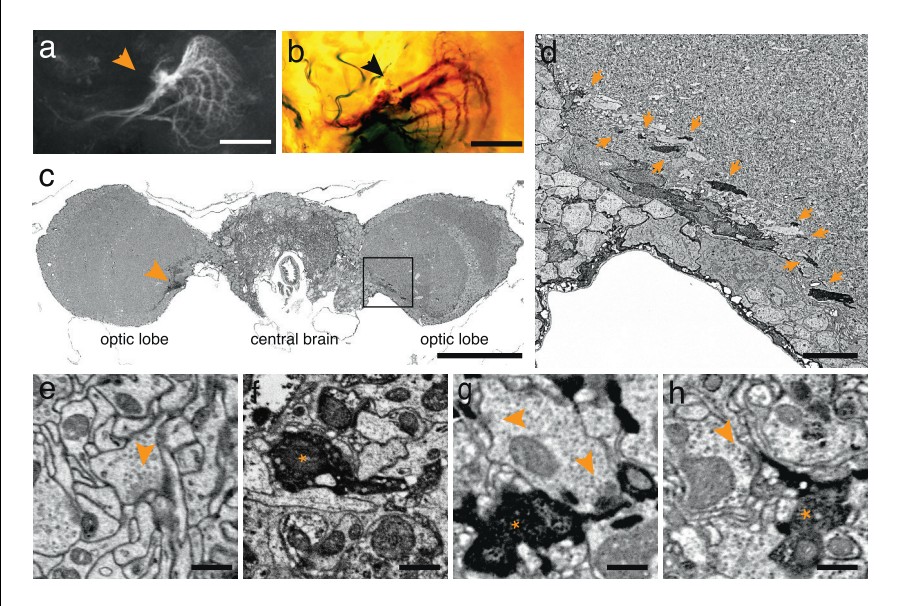

**Figure 2.** APEX expressing interneurons in *Drosophila melanogaster*. (**a**) Fluorescent image of a brain of *Drosophila melanogaster* (*DB331-Gal4 → UAS-APEX-GFP*) expressing APEX in direction-selective lobula plate tangential cells (LPTCs). (**b**) Bright field image of a similar plane from another (*DB331-Gal4 → UAS-APEX-GFP*) fly, labeled with DAB. (**c**) Electron micrograph of a frontal brain section (30 nm thickness) containing electron dense staining in axonal processes of LPTCs (arrowhead and framed box). (**d**) Enlarged view of the framed region in (**c**). Arrows point to electron dense processes. (**e**) Unstained synaptic terminal, showing quality of ultrastructural detail. (**f**) Synaptic nerve terminal in an APEX-positive process, identifiable by contrast-reversed vesicles. (**g,h**) APEX-positive postsynaptic processes, identifiable by the presence of adjacent vesicle-laden, T-bar-containing terminals. Scale bars: **a–c** :100 μm; **d** :10 μm; **e–g** : 500 nm.

stained neurites at overview resolution (20–30 nm per pixel), we tested the minimal requirements to reconstruct both cell types at these resolutions. Rapidly imaging every 10th section (270 nm separation), at 30 nm resolution was sufficient for mapping a J-RGC dendrite (144 sections covering $> 1 \times 10^7$ µm$^3$, imaged in 22 hr; *Figure 3a,b*). By comparison, imaging the same volume at high resolution (4 nm/pixel) would have taken ~2500 hr on the same microscope. Manual reconstruction of the J-RGC was straightforward and took 2 worker-hours (*Figure 3c*). To extend this result to another cell type, we imaged all 1260 sections from a bloc with APEX2NES-positive SACs and reconstructed

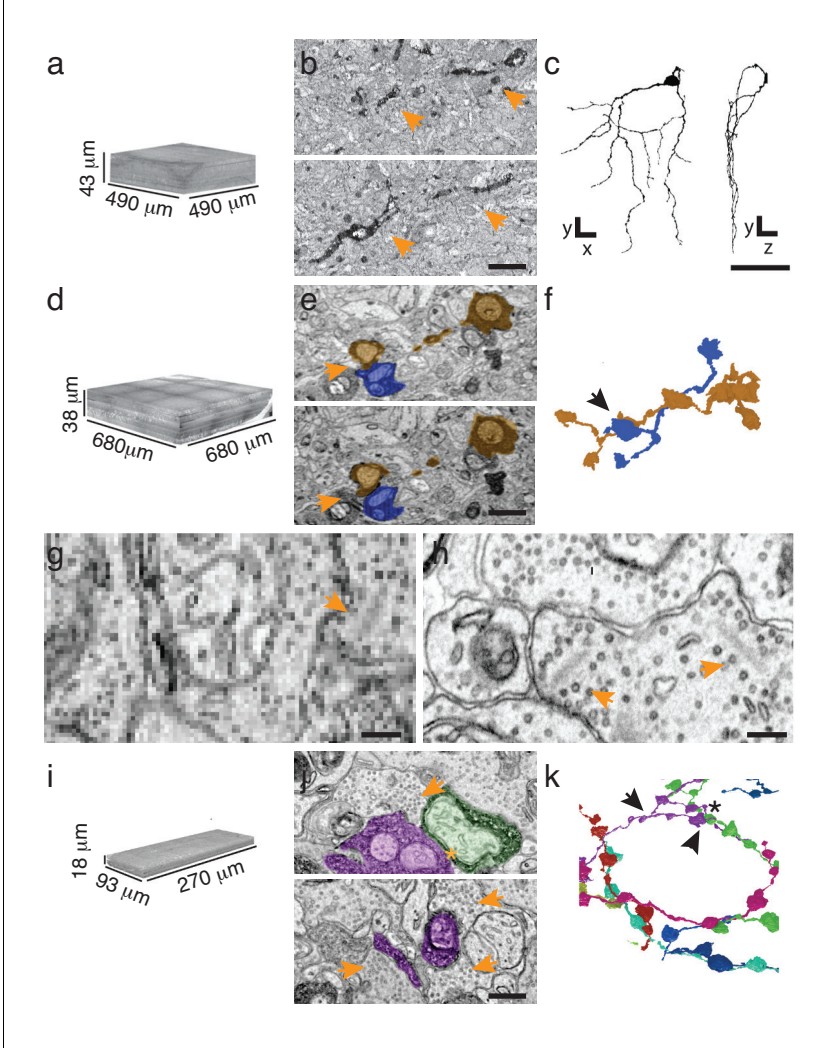

**Figure 3.** Targeted interrogation of serial-section EM volumes in mouse retina. (**a**). Overview-resolution EM volume containing J-RGCs expressing APEX2NES. Every 10th section was imaged at 30 nm/pixel. (**b**) Two consecutive sections containing APEX2NES expressing J-RGC processes (arrows). (**c**) Reconstructed J-RGC. Note the reduced complexity of the dendritic field due to the subsampling of the overview dataset. (**d**) Overview-resolution EM volume that contains SACs expressing APEX2NES. The sections were imaged at 20 nm/pixel. (**e**) Two consecutive sections showing a SAC-SAC contact (arrowheads). (**f**) Reconstruction of the SAC-SAC interaction in (**e**); arrowhead indicates contact shown in **e**. (**g–h**). Comparison of ultrastructural detail visible at 20 nm/pix (**g**) or 4 nm/pix (**h**). Although both images contain ribbon synapses, the low-resolution image (**g**) lacks the required resolution to identify them. Arrowheads point to ribbons. (**i**) High-resolution (4 nm/pixel) EM subvolume selected from (**d**). (**j**) Two high-resolution EM-micrographs containing SAC-processes expressing APEX2NES receiving synaptic contacts (arrowheads). (**k**) Reconstruction of the SAC plexus with its characteristic dendritic fasciculation and net-like structure. Asterisk indicates contact and arrowheads the synaptic inputs shown in **j** (top panel). Scale bars: **b**: 5 µm; **c**: 50 µm; **e**, **j**: 1 µm; **g–h**: 200 nm.

SAC processes in this volume (*Figure 3d–f*). Thus, we can reliably find and reconstruct genetically identified cells.

Although 20 nm resolution is sufficient for viewing neurites, it does not allow optimal visualization of synaptic contacts (*Figure 3g–h*). However, sections generated using ATUM are collected on wafers, and can be re-imaged; a program, Wafer-Mapper (*Hayworth et al., 2014*), facilitates returning to the same place on a given section with micron-level precision. This feature allows targeted imaging at high resolution of areas chosen from the rapidly acquired, lower resolution reconstruction, thereby substantially reducing imaging time. To test this multi-scale approach, we focused on the SACs, whose dendrites can be less than 100 nm in diameter. We selected a heavily-labeled ~5 × 10⁵ µm³ volume from the overview-resolution reconstruction of ~2 × 10⁷ µm³ and acquired a high-resolution (4 nm/pixel) data set in 250 hr (*Figure 3i*). Compared to a completely high-resolution approach, this amounts to a 40-fold reduction in imaging time. More importantly, it expedited reconstruction, the current time-limiting step in connectomics (*Plaza et al., 2014*), by constraining efforts to defined regions of interest. We manually traced and reconstructed the SAC plexus in this volume, observing multiple SAC-SAC interactions, dendritic fasciculation in the SAC plexus and dendritic branching (*Figure 3j,k* and data not shown).

Finally, to further expedite the reconstruction pipeline, we developed an algorithm that detects and segments APEX2-positive processes, taking advantage of the high contrast rendered by the DAB polymer. Our algorithm is fast, parameterized and does not rely on the supervised machine learning training regimes currently required for lower contrast material (*Kasthuri et al., 2015*) (*Figure 4a*, Materials and methods). We tested the algorithm using 401 sections from the high-resolution dataset (*Figure 3i*). We hand-segmented a set of 444 APEX2 positive segments, and then

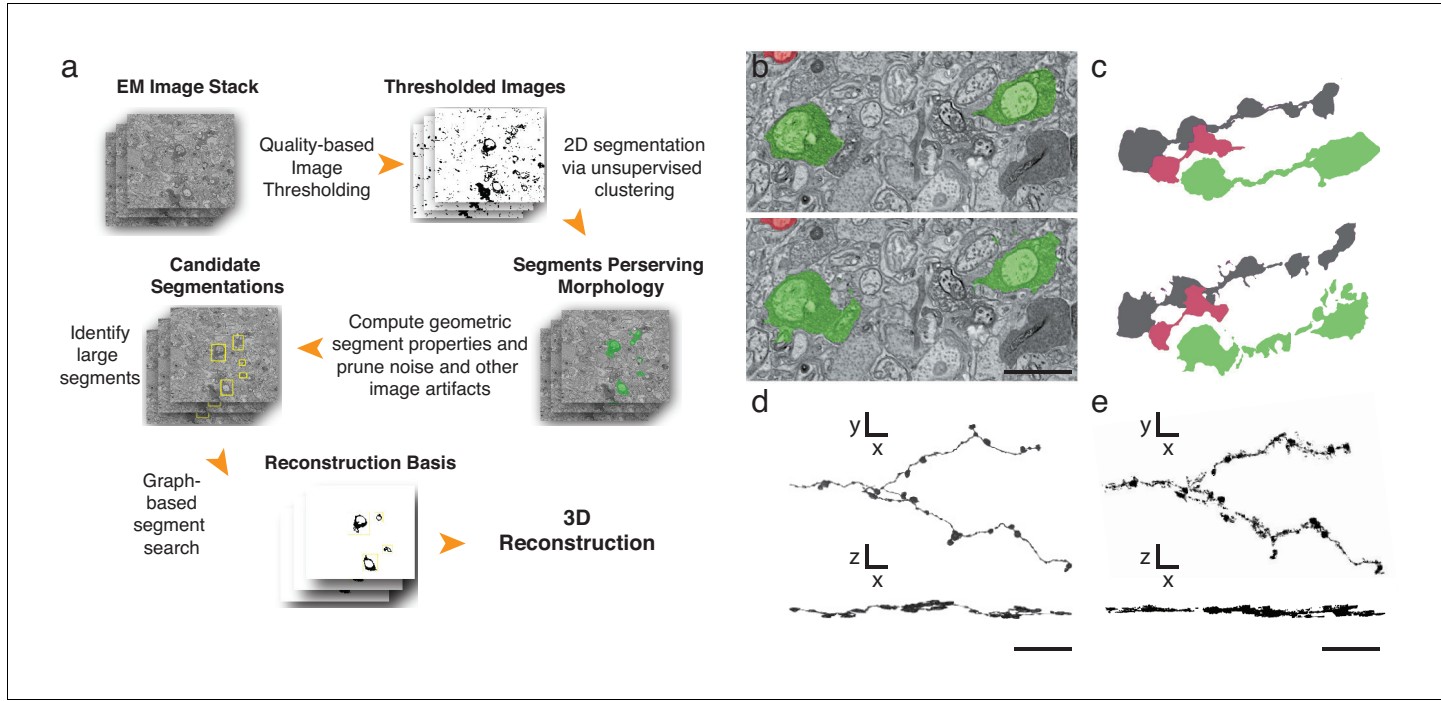

**Figure 4.** Automatic segmentation and reconstruction algorithm for APEX2 positive processes. (a) The algorithm performs unsupervised 2D segmentation and 3D reconstruction without the need of pre-training. The procedure follows these steps: **1**. Clustering-based image thresholding over pixel intensities from known contrast-enhanced ranges, modulated via quality assessment; **2**. 2D segmentation based on unsupervised clustering; **3**. Calculation of geometric properties for each identified segment, followed by noise and artifact pruning based on these properties; **4**. Identification of large segments based on segment properties; **5**. Graph-based segment search for reconstruction and labeling in 3D, followed by a final merging procedure to enforce consistency across the volume. (b) Unsupervised automatic segmentation of mouse SAC processes. Top: manual segmentation, ground-truth; bottom: automatic segmentation. (c).Reconstructions of the dendritic process in (b), comparing manual (top) and automatic reconstructions. (d) Manual reconstruction of a starburst amacrine cell process. *En face* (top) and side view (bottom). (e) Automatic reconstruction of the same process shown in d. *En face* (top) and side view (bottom). Scale bar in b: 1 µm; d, e: 10 µm.

compared these results to those obtained computationally. We calculated a recall statistic of 91.8% for the 2D segmentation portion of the algorithm (*Figure 4b,c*). All missed segments were small (on average 45 × 42 pixels), and were seldom necessary to establish connectivity. Consistent with human inferred segmentation, the analysis of segment adjacency in local neighborhoods can make connections in regions with small gaps. To test the accuracy of the algorithm for longer (>50 µm) processes, we segmented and reconstructed a dendritic branch of a starburst amacrine cell. Our algorithm could reconstruct the main characteristics of the dendritic processes and recapitulated most of the manually segmented details (*Figure 4d,e*). The primary advantage of this approach is processing time, being two orders of magnitude faster than approaches based on convolutional neural networks for membrane classification (*Kasthuri et al., 2015*; *Kaynig et al., 2015*). Training regimes in the neural network approaches require adjusting at least thousands of parameters separately for each new tissue; in contrast, our algorithm incorporates unsupervised components with a small fixed set of tunable parameters (see Methods and Supplementary Code). Importantly, the high contrast-to-noise ratio of the peroxidase-labeled cells is the enabling factor for the efficacy of this algorithm, which means that it could be used for reconstruction of DAB-stained structures in any tissue. Thus, although detailed synaptic properties may be obscured by the electron-dense stain, our methods are well-suited for rapid reconstruction of targeted neuron morphology and connectivity.

In summary, we have assembled a set of tools that enables rapid reconstruction of genetically identified neurons, so that their shapes and connections can be mapped at high resolution in much less time than required for conventional imaging and segmentation protocols. By expediting the analysis of neural motifs, ARTEMIS renders the interrogation of diverse samples feasible and holds a clear promise to unravel mechanisms ranging from neuronal development to computations.

## Materials and methods

### Animals

Animals were used in accordance with NIH guidelines and protocols approved by Institutional Animal Use and Care Committee at Harvard University. *JAM-B-creER* mice (*Kim et al., 2008*)were generated in our laboratory. *ChaT-cre* mice (*Rossi et al., 2011*), and *Cart-cre* mice (*Madisen et al., 2010*) were obtained from Jackson Laboratories. In the *Chat-cre* line, the Cre recombinase gene was targeted to the endogenous *Chat* gene; this line expresses Cre in starburst amacrine cells (SACs). In the *Cart-cre* transgenic line, Cre expression is controlled by regulatory elements from the *Cartpt* gene. In this line, Cre marks ON-OFF direction selective retinal ganglion cells. Mice were maintained on a C57/BL6J background. Both male and female mice were used in this study. Animals were 60 to 100 days old at the time of euthanasia.

Flies were raised on standard cornmeal-agar medium. The DB331 Gal4-line (*Joesch et al., 2008*) was kindly provided by Vivek Jayaraman (Janelia Research Campus) and the *UAS-APEX-GFP* (*Chen et al., 2015*) line was generated and kindly provided by Chiao-Lin Chen and Norbert Perrimon (Harvard Medical School and HHMI).

### Peroxidases

The endoplasmic reticulum-targeted HRP (erHRP) was designed as follows. First, to improve expression in mammals, the nucleotide sequence of HRP (horseradish peroxidase from a plant, Armoracia rusticana) was codon-optimized (DNA2.0, Menlo Park, CA). Second, to regulate protein trafficking a signal secretion sequence from the human immunoglobulin kappa chain (from pDisplay, Invitrogen, Carlsbad, CA) as well as an endoplasmic reticulum (ER)-retention signal (-KDEL) were appended at the N- and C-termini, respectively. Finally, the N175S mutation was introduced to confer heat stability and resistance to $H_2O_2$ (*Morawski et al., 2001*). The sequence of this cDNA is available from GenBank #KU504630. Ascorbate peroxidase (APX, dimeric) from pea (*Martell et al., 2012*) , and the enhanced monomeric version APEX2 (derived from soybean APEX) (*Lam et al., 2015*) were codon-optimized for better expression in mammals. In APEX2NES, the nuclear export signal (NES) was appended to APEX2. The erHRP, APX, and APEX2NES plasmids under CMV promoter were transfected to HEK293T cells (ATCC) using the DMRIE-C transfection reagent (Life Technologies). Subsequently, the cDNAs were cloned into a plasmid encoding a Cre-dependent Adeno-associated virus (AAV) backbone with the CAG (CMV-beta actin promoter + beta-globin leader) promoter,

woodchuck post-transcriptional element (WPRE), and the FLEX switch (*Atasoy et al., 2008*) (*Supplementary file 1*). AAV viruses were generated by transfecting these plasmids together with appropriate helper plasmids, and prepared using a chemical precipitation method (*Guo et al., 2012*).

Plasmids encoding the viral vectors will be sent to AddgenePlasmids described in this paper are available from Addgene (www.addgene.org).

## AAV-mediated gene transfer

For viral-mediated gene transfer, adult Cre-mice were anaesthetized with ketamine/xylazine by intraperitoneal injection. A 30 1/2G needle was used to make a small hole in the temporal eye, below the cornea. 1 μl of vitreous fluid was withdrawn and then 1 μl of rAAV2 or rAAV2/9 Cre-dependent viruses (a titre of ~1 × $10^{11-12}$ genome copies per ml) was injected into the subretinal space with a Hamilton syringe and 33G blunt-ended needle. Animals were euthanized and retinas were dissected 4–6 weeks following injection.

## Tissue preparation

Mouse retinas were dissected from eyecups in oxygenated Ames' medium (Sigma) with constant bubbling (95% $O_2$, 5% $CO_2$) at room temperature. Four incisions were made to flat-mount the retina with ganglion cells facing up onto nitrocellulose filter paper. The tissue was drop-fixed, with 2% PFA and 2.5% glutaraldehyde followed by 2.5% glutaraldehyde (times specified in *Supplementary file 2*) then washed. HEK-cells were fixed with 2% PFA and 2.5% glutaraldehyde for 15 min followed by a 45 min fix in 2.5% glutaraldehyde. Flies were decapitated and dissected in oxygenated Ringer solution. A small incision was made on the back of the head and the posterior cuticle was separated from the head. This ensured that the fixative and staining solutions could penetrate into the brain while the rest of the cuticle protected brain tissue during processing. Flies were fixed with 2% PFA and 2.5% glutaraldehyde for 15 min followed by a 45 min fix in 2.5% glutaraldehyde. Following aldehyde fixation, cells and tissues were washed, reacted with DAB to reveal sites of peroxidase activity, washed again, reduced with 50 mM sodium hydrosulfite and stained with osmium. Osmium treatments included 2% aqueous osmium tetroxide (used only in HEK cell micrographs of *Figure 1—figure supplement 1*) or the reduced osmium tetroxide-thiocarbohydrazide (TCH)-osmium ('rOTO') (*Willingham and Rutherford, 1984*; *Hua et al., 2015*; *Tapia et al., 2012*) protocol (all other Figures). Due to the two consecutive osmication steps, the 'rOTO' protocol improves the signal of membranes compared to the aqueous osmium. This enables reasonable signal-to-noise ratios at high scanning, an essential requirement for our approach. Finally, the stained tissue was dehydrated and infiltrated with Durcupan resin. Sodium-cacodylate (cat. no. 12300), glutaraldehyde (16220 and 16120), paraformaldehyde (15710), osmium tetraoxide (19190), maleic acid (18150), acetone (glass distilled; 10015) and uranyl acetate (22400) were purchased at Electron Microscopy Sciences (EMS); AMES medium (A1420), 3,3'-diaminobenzidine (DAB; D5905), potassium hexacyanoferrate (II) (P9387), thiocarbohydrazide (88535), sodium hydrosulfite (157953) and durcopan resin (44610) were purchased at Sigma-Aldrich. Concentrations and incubations times, along with details on reagents are provided in *Supplementary files 2* and *3* .

## Immunohistochemistry

For the image in *Figure 2a*, the fly brain was fixed in 2% paraformaldehyde in PBS for 30 min on ice, washed with PBS and blocked with 3% goat serum/1% Triton X-100/PBS. For staining, tissue was incubated with 3% goat serum/1% Triton X-100/PBS and rabbit anti-GFP Alexa Fluor 488 conjugate (dilution 1:1000, Invitrogen, A-21311) at 4°C for 1 days and washed with PBS. Brains were mounted on Vectashield mounting medium (Vectorlabs) and imaged in a confocal microscope (Olympus FVA). For the image in *Figure 2b*, brains were prepared for electron microscopic and imaged before osmication.

## Electron microscopy

The cured blocks were trimmed to a 2 × 3 mm rectangle and a depth of 400 μm and then readied for automated serial sectioning. The automated, unattended collection of ~ 30 nm serial sections was accomplished using a custom tape collection device (ATUM) (*Hayworth et al., 2014*) attached

to a commercial ultramicrotome. The sections were collected on plasma-treated carbon-coated poly-amide (Kapton, Sheldahl) 8-mm-wide tape. Sections were post-stained with 1% uranyl acetate in maleate buffer for 30 s and with 3% Lead Citrate (Ultrostain II; Leica - cat. no. 16707235) for 30 s. An automated protocol to locate and image sections on the wafers was used (*Hayworth et al., 2014*) with a Sigma scanning electron microscope (Carl Zeiss), equipped with the ATLAS software (Fibics). Images were acquired using secondary electron detection.

For the medium- and high-resolution data sets, non-affine alignment was accomplished through the FijiBento alignment package (https://github.com/Rhoana/FijiBento) that enables the alignment of large data sets in a relatively short period of time. To this end, the alignment was performed on the Odyssey cluster supported by the FAS Division of Science, Research Computing Group at Harvard University. The aligned images were then manually segmented using a custom volume annotation and segmentation tool (VAST; http://openconnecto.me/Kasthurietal2014/Code/VAST). The segmented images were processed for data analysis and 3D modeling with Matlab scripts, and Persistence of Vision Raytracer (http://www.povray.org/) for rendering steps.

## Automatic 2D segmentation and 3D reconstruction

Full resolution EM images ($100000 \times 50000$ pixels) underwent contrast adjustment via histogram equalization to normalize image intensities across slices. For each normalized image, a global threshold $\tau$ was computed via clustering-based image analysis (*Bankman, 2008*), modulated by the known contrast-enhanced pixel ranges of the ARTEMIS markers for a data set (e.g., for Suppl. *Figure 4b*: if $\tau < 0.6$, $\tau = \tau * 0.82$, else $\tau = \tau * 0.86$). The normalized images were then converted to a binary representation, and clustering-based image thresholding was applied again, assuming two classes (positive and negative), to gather 2D candidate segments. To remove artifacts such as speckle noise, and holes in the tissue, only segments that satisfy the following constraints were stored: 3 pixels < area < 130,000 pixels and major axis length < 900 pixels. The remaining segments were assigned a unique label, and stored in a MySQL database, along with their coordinates and other geometric properties Based on the expected morphological progression of each process, irregularly large imaging artifacts that appear between adjacent segments in the volume were pruned. Graph-based segment search over the database entries established the connectivity between segments, including those with gaps between them, by finding the minimum distance between centroid points from all pairs of segments in a local neighborhood using a k-Nearest Neighbors-like (*Hastie et al., 2009*) algorithm ($k = 2$). Using the graph as a guide, 2D segments were merged into the final 3D reconstruction. We included the source code in the supplementary information, packaged as a Matlab live script, with example images and an animation.

## Data

All relevant data is available on request. The electron microscopic data set of *Figure 3k*, which includes the segmented processes used for the reconstructions, has been deposited to Dyrad. Doi:10.5061/dryad.h67t6.

## Acknowledgements

We thank Alice Ting, Jeff Martell, and Stephanie Lam for advice and for providing the APX and APEX2 constructs. We also thank Chiao-Lin Chen and Norbert Perrimon for generating and providing the *UAS-APEX-GFP* fly and Benjamin de Bivort for support with the fly work. This work was supported by NIH grant NS76467 to MM, JL and JRS, an HHMI Collaborative Innovation Award to JRS, an IARPA contract #D16PC00002 to WJS and by The International Human Frontier Science Program Organization fellowship to MJ. The authors declare no conflicts of interest.

## Additional information

### Funding

| Funder | Grant reference number | Author |
| --- | --- | --- |
| NIH Blueprint for Neuroscience Research | R01NS076467 | Maximilian Joesch<br>David Mankus<br>Masahito Yamagata<br>Richard Schalek<br>Markus Meister<br>Jeff W. Lichtman<br>Joshua R. Sanes |
| Human Frontier Science Program | LT000167/2010 | Maximilian Joesch |
| Howard Hughes Medical Institute | HCIA program - 1234 | David Mankus<br>Masahito Yamagata<br>Joshua R. Sanes |
| Intelligence Advanced Research Projects Activity | #D16PC00002 | Walter J. Scheirer |

The funders had no role in study design, data collection and interpretation, or the decision to submit the work for publication.

### Author contributions

MJ, Conception and design, Acquisition of data, Analysis and interpretation of data, Drafting or revising the article; DM, Acquisition of data, Analysis and interpretation of data; MY, Conception and design, Acquisition of data, Contributed unpublished essential data or reagents; AS, Acquisition of data, Analysis and interpretation of data, Contributed unpublished essential data or reagents; RS, AS-P, Acquisition of data; MM, JWL, Conception and design; WJS, Conception and design, Analysis and interpretation of data; JRS, Conception and design, Analysis and interpretation of data, Drafting or revising the article

### Author ORCIDs

Masahito Yamagata, http://orcid.org/0000-0001-8193-2931
Markus Meister, http://orcid.org/0000-0003-2136-6506
Joshua R Sanes, http://orcid.org/0000-0001-8926-8836

### Ethics

Animal experimentation: Animals were used in accordance with NIH guidelines and protocols approved by Institutional Animal Use and Care Committee at Harvard University (Protocol 233 #92_19).

## Additional files

### Supplementary files

• Supplementary file 1. AAV constructs used in this study.

• Supplementary file 2. Fixation, reaction and staining method: rOTO protocol.

• Supplementary file 3. Fixation, reaction and staining method: reduced osmium protocol.

• Supplementary file 4. Table of contents - Artemis Live Scripts.mlx.

### Major datasets

The following dataset was generated:

| Author(s) | Year | Dataset title | Dataset URL | Database, license, and accessibility information |
|---|---|---|---|---|
| Joesch M, Mankus D, Yamagata M, Elia S, Shalek R, Suidda-Peleg A, Meister M, Licht-man J, Scheirer W, Sanes J | 2016 | Data from: Reconstruction of genetically identified neurons imaged by serial-section electron microscopy. | http://dx.doi.org/10.5061/dryad.h67t6 | Available at Dryad Digital Repository under a CC0 Public Domain Dedication |

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
