## [Decision Letter]

Thank you for submitting your article "Reconstruction of genetically identified neurons imaged by serial-section electron microscopy" for consideration by *eLife*. Your article has been reviewed by two peer reviewers, and the evaluation has been overseen by a Reviewing Editor and Eve Marder as the Senior Editor.

The reviewers have discussed the reviews with one another and the Reviewing Editor has drafted this decision to help you prepare a revised submission.

Summary:

Two complementary and timely advances are described for genetic tagging and reconstruction of subsets of retinal and *Drosophila* neurons by electron microscopy. Viral vectors are used that required Cre-dependent recombination and express erHRP, APX or APEX2. These were tested in mouse retinas, in which certain retinal ganglion cells expressed Cre-recombinase, or tamoxifen activated Cre, and also in *Drosophila* that expressed APEX fused to GFP in direction selective tangential cells. Listing the specimen preparation methods that led to the success or failure of APEX and HRP-dependent staining will benefit the field. Likewise, the combined application of rapid low resolution with selected high-resolution analysis will facilitate the time-consuming, but much-needed large volume reconstructions of genetically specified neurons by serial electron microscopy. The paper also describes other aspects of a workflow to reconstruct the labeled cells. These approaches are described as being quicker than other current methods. The entire method has considerable potential and will be of interest to the neuroscience community. Despite this overall favorable assessment of the study, several points should be addressed.

Essential revisions:

1) An important question concerns efficiency. Are all of the genetically specified neurons labeled or just a subset? For example, the *Drosophila* DB331 driver line should target a bundle of motion-sensitive cells with numerous spatially constrained projections, yet Figure 2 displays only a handful of processes. In both Figure 1 (C-D) and Figure 2, a low-resolution comparison of Cre-dependent fluorophore and DAB staining patterns are needed to address this concern.

2) The paper does a convincing job of showing how 'rapid' low-resolution images and reconstructions can be made of axons and dendrites. Nevertheless, to validate the method's use in circuit analysis, consistent high quality images are needed that show synapses identified by arrows with detailed descriptions in the figure legends and results (a few of the images are of sufficient magnification and quality but synapses should be identifiable (Figure 3 and Figure 1—figure supplement 3 and Figure 1—figure supplement 4)). Figure 1—figure supplement 3 is the only image that actually identifies the synapses with arrows, yet still provides too little description.

3) All of the example images reveal suboptimal ultrastructure inside the labelled neurites, relative to other methods that have been used to label synapses in the literature. The dense reaction product obscures pre- and postsynaptic active zones, as well as presynaptic vesicles. This being the case, it is not clear how synapses will be identified on the specifically labeled cells. This issue must be addressed with images, labels, and clear examples of labeled synapses.

4) Both aqueous osmium and reduced osmium (rOTO protocol) were used. However, it is not completely clear which osmium staining was used in some of the figures. In addition, it is not entirely clear whether there were any significant differences between the single osmium or rOTO protocol methods for low magnification reconstructions or synapse identifications. The comparisons in Figure 1—figure supplement 1 need more explanation.

5) It appears that the findings are intended to be qualitative; however, some form of semi-quantitative comparisons are warranted to establish the claims. For example, blind scoring of synapses (such as 0-3) and ultrastructural quality under the different conditions would be welcomed to assess the stated improvements.

6) Similarly, a systematic (and blind as to condition) quantification of a subset of small volumes of synapses from control and reacted tissue is needed to rule out possible effects of peroxidase expression on the ability to identify synapses in either the labelled cells or among the associated or surrounding but unlabeled synapses.

7) The authors indicate that the technique provides improved preservation. Although in Figure 1, panels H to L are of acceptable magnification and resolution, they do not show particular evidence of improved preservation relative to other papers in the literature. Description and discussion are needed to clarify to what the improvement has been made.

8) It is noted that the fixation of the fly brain was with immersion, without dissecting it away from the cuticle. Only an incision was made in the back of the head. Some explanation is needed as to why the authors used this approach, instead of removing the whole cuticle as is standard in the literature. The tissue preservation does not look as good as that in other published images where the whole cuticle was removed, and fixative penetration was facilitated, for example with microwave irradiation.

9) Concerning labelling in the fly brain, the outcomes should be described in more detail in the Results section (Results and Discussion, fifth paragraph). Higher magnification images are needed in Figure 2 to identify synapses, which should be labelled on the images and described in the Figure legend. These higher resolution images could replace one or more of the panels in Figure 2. It is important to show here the synaptic contacts with the labelling on either the pre or post-synaptic side. We recommend that sufficient magnification be shown so that synapses are readily identifiable, with arrows, and more detailed descriptions in the results and legends.

10) The description of the automatic 2D segmentation and 3D reconstruction explains that speckle noise is removed by pruning connected components smaller than 1500 pixels in area. As written, it is not clear how this removal was done. Based on the low resolution that was used for the EM imaging, the size of the removed component seems rather large. Description of how effective this process was when dealing with long thin neurites is needed.

11) A more direct statement is needed in Figure 3 legend, that 20 nm resolution is not sufficient to identify synapses, even in the unlabeled processes. Then in Figure 3 (4 nm resolution), synapses should be explicitly identified with arrows.

12) The serial sections selected for Figure 3 should show a synapse with one of the labelled cells, which is needed to validate the circuit reconstruction.

13) [Supplementary-material SD1-data]–[Supplementary-material SD3-data] were not included in the PDF for review and need to be submitted with the revision for proper review.

14) In addition, it would be helpful if the software could be prepared for expert review upon revision.

[Editors' note: further revisions were requested prior to acceptance, as described below.]

Thank you for resubmitting your work entitled "Reconstruction of genetically identified neurons imaged by serial-section electron microscopy" for further consideration at *eLife*. Your revised article has been favorably evaluated by Eve Marder (Senior editor), a Reviewing editor, and two reviewers.

In general, the authors have addressed prior concerns raised by the reviewers. There remain a few outstanding issues described below.

Essential further revisions:

1) Issue of overstatements on 'improved ultrastructure' needing more work:

A) Abstract – remove the phrase "and improves ultrastructural preservation". This method does not improve ultrastructural preservation; it enhances staining. Could substitute "while retaining ultrastructure" or something similar. The purpose of this method is not to improve ultrastructural preservation, and as noted in the prior review, the ultrastructure of the labeled cells is not particularly good with respect to other methods.

B) Introduction – second paragraph – (b) enhancing the electron-density of the stain without compromising ultrastructure. To be clear reword as "without compromising the surrounding ultrastructure" or in a similar way. As written, it sounds like the ultrastructure of the labelled cell would not be compromised, but of course it is compromised as the stain obscures most of the ultrastructure.

C) Consider the following concern raised by one of the reviewers and revise the Introduction and Discussion accordingly:

The narrative does not oversell the technique applicability for ultrastructure analysis. However, since the ability to examine synaptic ultrastructure in genetically designated neurons would represent a significant milestone, it may be more appropriate to characterize explicitly this current incarnation of peroxidase-based methods as most appropriate for rapid reconstruction of targeted neuron morphology and connectivity, rather than synapse properties.

2) Some specific figures still need to be improved:

A) Figure 2: The addition of panels showing complementary GFP and DAB staining is helpful. However, since a single *APEX-GFP* construct is used, the authors should address the differences between panels A and B either in Results or the figure legend. Specifically, while DAB produces symmetrical staining, GFP is quite a bit stronger in one lobe; furthermore, prominent puncta are visible in panel A that have no match in panel B. What are the puncta? Can the authors confirm that 2A is indeed GFP, despite the puncta and the general graininess of the image? Finally, Figure 2 legend is vague as to whether or not a section from the brain in 2A is shown following DAB polymerization – it probably is not the same brain, and this should be made clear.

B) Figure 1 and Figure 2 are not particularly convincing images: If these are from a series, then perhaps showing the subsequent images in a supplemental figure would help convince a reader about the nature of the synaptic contact.

C) Provide new Figure 1—figure supplement 1 – these were mentioned before, and un-necessarily make the standard look worse due to folds and knife marks in the images.

3) In each figure legend, state explicitly the mouse or fly sources of the images.

---

## [Author Response]

Essential revisions:

1) An important question concerns efficiency. Are all of the genetically specified neurons labeled or just a subset? For example, the Drosophila DB331 driver line should target a bundle of motion-sensitive cells with numerous spatially constrained projections, yet Figure 2 displays only a handful of processes. In both Figure 1(C-D) and Figure 2, a low-resolution comparison of Cre-dependent fluorophore and DAB staining patterns are needed to address this concern.

As the reviewers noted, the DB331-Gal4 line expresses Gal4 in a set of motion-sensitive cells comprising 6 vertically and 3 horizontal motion sensitive lobula plate tangential cells, plus a sparse group of small interneurons. Confusion arose because in the original Figure 2, the plane of section revealed the axonal, but not the dendritic processes, which are posterior to the axons. Indeed, more posterior sections do show the expected complex pattern of dendritic processes expected from these tangential cells (data not shown). We have taken two steps to address the concern. First, we show a section in which 9 DAB-positive processes, the expected number, are visible. We have added arrows to point them out. Second, as suggested, we have added low- resolution panels to compare the Gal4 dependent expression of fluorophore (*APEX*-GFP) and DAB reaction product (Figure 2). These micrographs document the correspondence between these two reporters.

2) The paper does a convincing job of showing how 'rapid' low-resolution images and reconstructions can be made of axons and dendrites. Nevertheless, to validate the method's use in circuit analysis, consistent high quality images are needed that show synapses identified by arrows with detailed descriptions in the figure legends and results (a few of the images are of sufficient magnification and quality but synapses should be identifiable (Figure 3 and Figure 1—figure supplement 3 and Figure 1—figure supplement 4)). Figure 1—figure supplement 3 is the only image that actually identifies the synapses with arrows, yet still provides too little description.

We have added several high-resolution images in which synapses (pointed out with arrows) can be identified. They are Figure 1; Figure 1—figure supplement 4; and Figure 1—figure supplement 5. We have also provided more detailed descriptions in the text and figure legends.

3) All of the example images reveal suboptimal ultrastructure inside the labelled neurites, relative to other methods that have been used to label synapses in the literature. The dense reaction product obscures pre- and postsynaptic active zones, as well as presynaptic vesicles. This being the case, it is not clear how synapses will be identified on the specifically labeled cells. This issue must be addressed with images, labels, and clear examples of labeled synapses.

As the reviewers note, the cytosolic DAB-polymer deposited in APEX-expressing neurons obscures the ultrastructure of the labeled neurites. Nonetheless, key aspects of synaptic interactions can still be clearly identified. For example, cells presynaptic to APEX-expressing cells have unaffected synaptic machinery, including presynaptic densities, ribbons and vesicles. In addition, even within APEX-positive cells, synaptic vesicles can be visualized because they remain unstained. Moreover, erHRP is confined to membrane bound compartments, leaving the cytosol unstained; in cells expressing erHRP, both pre- and postsynaptic structures can be visualized. To emphasize these points, we have provided new images in Figure 1, and O andalso added a new supplementary figure, Figure 1—figure supplement 5. These figures present images of pre- and postsynaptic partners to APEX2 positive cells in the retina and superior colliculus, as well as synapses onto erHRP expressing retinal ganglion cells. We have also revised the text and figure legend to emphasize that ultrastructural detail preserved in peroxidase expressing cells does allow identification of pre- and postsynaptic partners. That said, previous publications make clear that postsynaptic densities are not as prominent in the retina as in other brain regions such as cortex. We also note this in the text.

4) Both aqueous osmium and reduced osmium (rOTO protocol) were used. However, it is not completely clear which osmium staining was used in some of the figures. In addition, it is not entirely clear whether there were any significant differences between the single osmium or rOTO protocol methods for low magnification reconstructions or synapse identifications. The comparisons in Figure 1—figure supplement 1 need more explanation.

We used the “rOTO” protocol in all experiments on retinal tissue, and now include a sentence in the Materials and methods section clarifying this. This protocol improves the signal strength of membranes, dramatically improving the signal-to-noise ratio at short pixel dwell times, thereby allowing increased scanning speed. We used the single osmium protocol only in the HEK cell experiment in Figure 1—figure supplement 1, to provide additional evidence on the importance of the redox state of the reaction products. We edited the text to clarify this point.

5) It appears that the findings are intended to be qualitative; however, some form of semi-quantitative comparisons are warranted to establish the claims. For example, blind scoring of synapses (such as 0-3) and ultrastructural quality under the different conditions would be welcomed to assess the stated improvements.

Figure 1—figure supplement 4 was intended to show the improvements of the “signal-to-noise” ratio between the membrane and cytosolic when comparing a standard with a tissue reduction protocol. We have now added high-resolution images of representative synapses to this Figure. As an additional “semi-quantitative” comparison, we asked a blind-to-condition observer to select the better example between pairs of manually selected images of synapses (images as in Figure 1—figure supplement 1). This randomized pairwise comparison shows that on average, the reduction improves synaptic recognition by 40% by reducing the tissue. This seems to be because presynaptic densities are more prominent in the reduced condition. Results of this comparison are now presented in Figure 1—figure supplement 4.

6) Similarly, a systematic (and blind as to condition) quantification of a subset of small volumes of synapses from control and reacted tissue is needed to rule out possible effects of peroxidase expression on the ability to identify synapses in either the labelled cells or among the associated or surrounding but unlabeled synapses.

As discussed in our response to point #3, pre- and postsynaptic partners can be distinguished and characterized in peroxidase expressing neurons. Qualitatively, these synaptic connections do not seem to vary in APEX2 or erHRP expressing cells, compared to control tissue. Although we agree that an unbiased (blind to condition) comparison would be valuable, it is not feasible. Due to the prominence the peroxidase staining it is also not trivial to design an experiment that is blind to condition. In addition, the size and prominence of synaptic specializations varies markedly among retinal subtypes. For example, ribbons in rod bipolar cells are larger and more pronounced than those in cone bipolars. Therefore, cell type specific markers, which we did not use, would be needed for valid comparisons. As an approximation, we noted the number of synapses that we counted per unit area in peroxidase-expressing and control retinas. We identified 0.34 synapses per μm2 in control tissue 0.32 synapses per μm2 in peroxidase expressing tissue. The difference was not significant. We have added this comparison to the text.

7) The authors indicate that the technique provides improved preservation. Although in Figure 1, panels H to L are of acceptable magnification and resolution, they do not show particular evidence of improved preservation relative to other papers in the literature. Description and discussion are needed to clarify to what the improvement has been made.

We apologize for being unclear in the text. We did not intend to claim that we have the best preservation relative to other papers in the literature. High pressure freezing, for example, may avoid the tissue distortions of chemical fixation protocols. Our claim is that our reduction protocol improves the contrast difference between membranes and cytosol by comparison to conventionally fixed tissue, and also enhances the signal of the presynaptic machinery (Please also see our response to point #5). We believe that these effects might be important for automatic segmentation algorithms that are expected to segment large data sets in the future. We have revised the text and figure legend (Figure 1—figure supplement 4) to avoid confusion.

8) It is noted that the fixation of the fly brain was with immersion, without dissecting it away from the cuticle. Only an incision was made in the back of the head. Some explanation is needed as to why the authors used this approach, instead of removing the whole cuticle as is standard in the literature. The tissue preservation does not look as good as that in other published images where the whole cuticle was removed, and fixative penetration was facilitated, for example with microwave irradiation.

We apologize for an incomplete description of the method, and have expanded the description. After making the incision we separate the posterior cuticle from the head. This allows unhindered access of the fixative to the brain tissue. We reasoned that by completely removing the cuticle we might do more damage to the tissue. The remaining cuticle was left intact to protect the tissue during all the staining procedure. We agree with the reviewers that this is not the best fly brain preservation. Our protocol was optimized for mammalian tissue. Our intention here was to test the generality of the method by determining whether it could be applied to flies; we show than it can. There is a clear path of improvement regarding tissue preservation in flies. We clarified this in the text.

9) Concerning labelling in the fly brain, the outcomes should be described in more detail in the Results section (Results and Discussion, fifth paragraph). Higher magnification images are needed in Figure 2 to identify synapses, which should be labelled on the images and described in the Figure legend. These higher resolution images could replace one or more of the panels in Figure 2. It is important to show here the synaptic contacts with the labelling on either the pre or post-synaptic side. We recommend that sufficient magnification be shown so that synapses are readily identifiable, with arrows, and more detailed descriptions in the results and legends.

As requested by the reviewers, we replaced panels in Figure 2 with high magnification images. One shows a presynaptic contact to an APEX-positive cell, a second one of an APEX- positive cell containing a number of contrast-inverted vesicles, similar to those observed APEX- positive cells in the superior colliculus. As described in our response to point #8, our protocol was optimized for mammalian retina tissue and we acknowledge that there is need of improvement in the preservation protocol of the fly brain. We make this point clear in the text.

10) The description of the automatic 2D segmentation and 3D reconstruction explains that speckle noise is removed by pruning connected components smaller than 1500 pixels in area. As written, it is not clear how this removal was done. Based on the low resolution that was used for the EM imaging, the size of the removed component seems rather large. Description of how effective this process was when dealing with long thin neurites is needed.

The algorithm’s parameters are set based on the regions under consideration from a specific data set (i.e., the histogram profile of the pixel intensities in that region). We have described the pruning process for noise and other artifacts in the Materials and methods section of the supplemental material (it incorporates segment area and major access length), and have provided an in-depth description of the algorithm’s reconstruction methodology in the attached Matlab live script walk-through.

*11) A more direct statement is needed in Figure 3 legend, that 20 nm resolution is not sufficient to identify synapses, even in the unlabeled processes. Then in Figure 3 (4 nm resolution), synapses should be explicitly identified with arrows.*

We added the following sentence: “Although both images contain ribbon synapses, the low resolution image (G) lack the required resolution. Arrows point to the location of synaptic ribbons.”

12) The serial sections selected for Figure 3 should show a synapse with one of the labelled cells, which is needed to validate the circuit reconstruction.

As noted above, synaptic specializations cannot be visualized when both presynaptic and postsynaptic cells have DAB-polymer in the cytosol. We base identification of the synapses between APEX-positive starburst amacrine cells (Figure 3) on the morphological criteria used Briggman et al. (2011) who faced a similar problem owing to the staining protocol they used. We can, however, visualize all other unstained synaptic partners based on their ultrastructural features. We replaced Figure 3 with high magnification images showing additional synaptic contacts to the reconstructed cells and pointed these out with arrowheads.

13) [Supplementary-material SD1-data]–[Supplementary-material SD3-data] were not included in the PDF for review and need to be submitted with the revision for proper review.

The files were uploaded but were unfortunately omitted from the final PDF provided to reviewers. We apologize for failing to notice this when we approved the PDF. These supplementary files are now included.

*14) In addition, it would be helpful if the software could be prepared for expert review upon revision.*

We now include a step-by-step Matlab script with examples as supplementary information.

[Editors' note: further revisions were requested prior to acceptance, as described below.]

Essential further revisions:

*1) Issue of overstatements on 'improved ultrastructure' needing more work:*

*A) Abstract – remove the phrase "and improves ultrastructural preservation". This method does not improve ultrastructural preservation; it enhances staining. Could substitute "while retaining ultrastructure" or something similar. The purpose of this method is not to improve ultrastructural preservation, and as noted in the prior review, the ultrastructure of the labeled cells is not particularly good with respect to other methods.*

Reworded as requested.

*B) Introduction – second paragraph – (b) enhancing the electron-density of the stain without compromising ultrastructure. To be clear reword as "without compromising the surrounding ultrastructure" or in a similar way. As written, it sounds like the ultrastructure of the labelled cell would not be compromised, but of course it is compromised as the stain obscures most of the ultrastructure.*

Reworded as requested.

*C) Consider the following concern raised by one of the reviewers and revise the Introduction and Discussion accordingly:*

The narrative does not oversell the technique applicability for ultrastructure analysis. However, since the ability to examine synaptic ultrastructure in genetically designated neurons would represent a significant milestone, it may be more appropriate to characterize explicitly this current incarnation of peroxidase-based methods as most appropriate for rapid reconstruction of targeted neuron morphology and connectivity, rather than synapse properties.

We have included this limitation as requested.

*2) Some specific figures still need to be improved:*

*A) Figure 2: The addition of panels showing complementary GFP and DAB staining is helpful. However, since a single APEX-GFP construct is used, the authors should address the differences between panels A and B either in Results or the figure legend. Specifically, while DAB produces symmetrical staining, GFP is quite a bit stronger in one lobe; furthermore, prominent puncta are visible in panel A that have no match in panel B. What are the puncta? Can the authors confirm that 2A is indeed GFP, despite the puncta and the general graininess of the image? Finally, Figure 2 legend is vague as to whether or not a section from the brain in 2A is shown following DAB polymerization – it probably is not the same brain, and this should be made clear.*

We confirm that 2A does show GFP. The signal was enhanced by immunostaining with anti-GFP, which is our lab’s standard protocol. The puncta likely reflect antibody background rather than ectopic GFP.

As to the differences between A and B, they reflect the differences in tissue preparation and staining protocol rather than differences in which cells are labeled. We ascribe the lack of bilateral symmetry in A to imperfections in tissue clearing and preparation; it is not a consistent finding. By showing both panels at higher magnification, we emphasize the similarity.

The sections in A and B are from different animals. We have added this information to the legend.

*B) Figure 1 and Figure 2 are not particularly convincing images: If these are from a series, then perhaps showing the subsequent images in a supplemental figure would help convince a reader about the nature of the synaptic contact.*

We substituted better micrographs for Figure 1 and Figure 2. In Figure 2, we now include examples of both pre- and postsynaptic APEX-positive processes, as well as an unstained synapse from this material to demonstrate the quality of the ultrastructure.

*C) Provide new Figure 1—figure supplement 1 – these were mentioned before, and un-necessarily make the standard look worse due to folds and knife marks in the images.*

We have substituted new images for supplemental Figure 1. 1C has no folds and 1E has far fewer folds than its predecessor. Since this is an issue of aesthetics rather than clarity, we do not believe it is worth spending more time preparing new tissue.

*3) In each figure legend, state explicitly the mouse or fly sources of the images.*

Added as requested.